# Identity-Driven Multimedia Forgery Detection via Reference Assistance

## ABSTRACT

Recent technological advancements, such as the "deepfake" techniques, have paved the way for generating various media forgeries. In response to the potential hazards of these media forgeries, many researchers engage in exploring detection methods, increasing the demand for high-quality media forgery datasets. Despite this, existing datasets have certain limitations. Firstly, most datasets focus on manipulating visual modality and usually lack diversity, as only a few forgery approaches are considered. Secondly, the quality of media is often inadequate in clarity and naturalness. Meanwhile, the size of the dataset is also limited. Thirdly, it is commonly observed that real-world forgeries are motivated by identity, yet the identity information of the individuals portrayed in these forgeries within existing datasets remains under-explored. For detection, identity information could be an essential clue to boost performance. Moreover, official media concerning relevant identities on the Internet can serve as prior knowledge, aiding both the audience and forgery detectors in determining the true identity. Therefore, we propose an identity-driven multimedia forgery dataset, IDForge, which contains 249, 138 video shots. All video shots are sourced from 324 wild videos of 54 celebrities collected from the Internet. The fake video shots involve 9 types of manipulation across visual, audio, and textual modalities. Additionally, IDForge provides extra 214, 438 real video shots as a reference set for the 54 celebrities. Correspondingly, we design an effective multimedia detection network termed the Reference-assisted Multimodal Forgery Detection Network (R-MFDN). Through extensive experiments on the proposed dataset, we demonstrate the effectiveness of R-MFDN on the multimedia detection task.

## CCS CONCEPTS

• **Computing methodologies → Artificial intelligence**.

## KEYWORDS

forgery detection dataset, identity information, forgery detector

## 1 INTRODUCTION

Recent advancements in deepfake techniques have provided users with tools for manipulating the content of individuals in videos or images by altering their facial features. However, the scope of such manipulation in the community has been extended beyond mere

*ACM MM, 2024, Melbourne, Australia*
© 2024 Copyright held by the owner/author(s). Publication rights licensed to ACM.
ACM ISBN 978-x-xxxx-xxxx-x/YY/MM
https://doi.org/10.1145/nnnnnnn.nnnnnnn

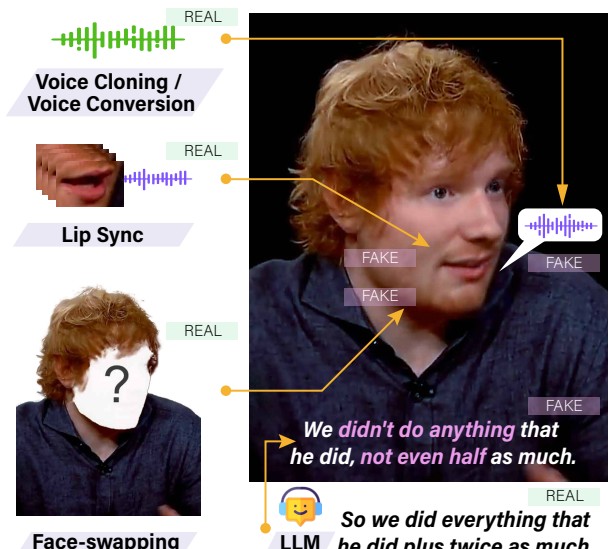

**Figure 1: The proposed IDForge dataset involves manipulation across modalities to create false identities, including techniques such as text manipulation, voice cloning, lip-syncing, and face-swapping, etc.**

visual alteration. Voice cloning and text-to-speech techniques are commonly employed at the audio level to generate vocal utterances with given tones. In addition, at the semantic level, large language models (LLMs) like GPT-4 [39] can be misused to generate content that closely mimics the expressive style of a target person.

As the threat posed by media forgeries continues to grow, many media forgery detection methods have emerged as countermeasures. The effectiveness of these approaches largely depends on high-quality media forgery datasets. However, current datasets have limitations that hinder progress in media forgery detection. Firstly, most current datasets focus on a limited range of forgery methods, primarily concentrating on visual manipulation. This narrow focus restricts detection methods to these specific types of manipulation, making them vulnerable to other forms of forgery that make manipulations on multiple modalities. Secondly, although there are a few datasets that incorporate manipulation across modalities, these datasets often exhibit noticeable deficiencies in the quality of forgeries. When compared to meticulously crafted forgeries found on the internet, the shortcomings, particularly in the quality of fake audio, become evident. This includes issues related to tone, clarity, and naturalness, undermining the utility of multimodal information for detecting forgeries. Thirdly, most datasets are not in an identity-driven design, failing to include identity-related information, i.e., extra reference media of given individuals. While some datasets contain a substantial number of video shots, these media are distributed among numerous individuals, each contributing

only a limited number of video shots. Notably, identity information frequently plays a pivotal role, particularly as numerous cases of media forgery seek to fabricate false identities. However, the scarcity of reference material and the restricted amount of media attributable to each individual impede detectors' capacity to compare and harness identities effectively for detection purposes.

Identity forgery, which refers to the deliberate manipulation of individuals' portrayed identities in media, constitutes a substantial portion of media forgeries. In identity forgery, information from various modalities, such as video content, audio sequences, and corresponding spoken content, is often combined to create a convincing false identity. Celebrities are prime targets due to their abundant publicly accessible media resources, making individual-targeted manipulation easier. Their fame amplifies the impact of such forgeries. For example, a gaming video featuring President Joe Biden appearing in the virtual game world of Skyrim is uploaded on YouTube. The face, voice, and spoken content are entirely fabricated and synchronized in this video. The fake identity of Joe Biden appears realistic and humorous, attracting millions of online views.

From the perspective of forgery detection, the abundant online media resources for celebrities emerge as a significant source of prior knowledge. It is intuitive to anticipate that when confronted with videos exhibiting potential signs of forgery, viewers will naturally draw upon their recollections of the celebrity in memory or seek additional information online. Viewers can quickly determine the video's authenticity by comparing the video's identity information with the corresponding reference.

Motivated by this, this work delves into identity-driven multimedia forgery detection, aiming to unravel the role of identity information in multimedia forgery detection. Given the absence of any existing dataset that furnishes comprehensive and accurate identity information, we commence by creating a high-quality identity-driven multimedia forgery dataset - IDForge, as illustrated in Figure 1. The proposed IDForge contains 249, 138 high-quality video shots. These video shots originate from 324 full-length YouTube videos covering 54 English-speaking celebrities. IDForge involves 9 types of visual, audio, and textual manipulations. We employ state-of-the-art manipulation methods to obtain natural, realistic, high-quality forgery video shots and manually select similar lookalikes for given identities during face-swapping. In total, we have 169, 311 forgery video shots. Unlike existing datasets, IDForge contains an additional reference set that provides abundant identity information for each celebrity. The reference set contains 214, 438 pristine video shots sourced from another 926 full-length YouTube videos.

Accordingly, we propose a Reference-assisted Multimodal Forgery Detection Network (R-MFDN) that introduces abundant identity information as a reference through identity-aware contrastive learning. Given suspected media data for one identity, Indentity-aware contrastive learning aims to learn identity-aware features for each modality by contrasting its feature with that from non-matching identities. In addition, to take advantage of multimodal information and mine inconsistency from different modalities for forgery detection, our R-MFDN introduces cross-modal contrastive learning to contrast features from different modalities. With cross-modal contrastive learning and identity-aware contrastive learning, R-MFDN can capture the inconsistency between different modalities

and leverage abundant identity information to improve the performance of multimedia forgery detection. The contributions of our works can be summarized as follows:

- We propose an identity-driven multimedia forgery dataset - IDForge, which contains 463, 576 high-quality video shots and is currently the largest multimedia forgery dataset.
- We propose the Reference-assisted Multimodal Forgery Detection Network (R-MFDN), which utilizes abundant identity information and mines cross-modal inconsistency for forgery detection.
- We conduct extensive experiments to demonstrate that our IDForge dataset is challenging and our proposed R-MFDN model outperforms the state-of-the-art baselines in media forgery detection.

## 2 RELATED WORK

### 2.1 Identity Forgery

Identity forgery in the online community mainly focuses on visual and audio aspects, i.e., deepfake videos and deepfake voices. Most of the deepfake videos contain realistic but synthetic faces. Many open-source toolkits manipulate faces by swapping them with another person's face, e.g., Faceswap-GAN [36], FaceSwap [49], and DeepFaceLab [40]. In addition, another type of deepfake video uses talking face generation methods to synthesize lip movements based on given audio, such as Wav2Lip [41]. Most deepfake video methods are based on Generative Adversarial Networks (GANs) [20], which employ an encoder-decoder architecture with one encoder and two decoders. Specifically, the encoder is trained to learn the common features of the source and the target faces, while the two decoders are trained separately to generate the source and target faces. Deepfake voice, also known as voice cloning, mimics someone's voice with a high degree of accuracy. There are also several open-source toolkits [2, 11, 51] for deepfake voice. Most of them are based on the SV2TTS framework proposed by Jia *et al.* [27]. The common generation process involves a speaker encoder that derives an embedding from a short utterance of the target speaker. Then, a synthesizer, conditioned on the embedding, generates a spectrogram from the given text. Finally, a vocoder is used to infer an audio waveform from the spectrograms generated by the synthesizer.

Combining deepfake video and voice cloning techniques can lead to highly realistic and convincing false identities of target individuals. However, most existing media forgery datasets only involve deepfake videos, limiting their ability to represent real-world forgery scenarios. In contrast, the media forgeries in our IDForge dataset include both deepfake video and voice generation methods, enabling a high-quality multimodal dataset.

### 2.2 Media Forgery Detection

Most media forgery detection methods [21, 24, 32, 33, 35, 37, 42, 58–61, 63] primarily identify media forgery by detecting visual defects in deepfakes. With the rapid development of forgery detection techniques, more recent methods have trended towards utilizing multimodal information for media forgery detection. For example, Zhou et al. [62] proposed a method that detects forgery by jointly

Table 1: Comparision of existing media forgery datasets. *Ref* refers to the reference set.

| Dataset | Date | Real Video Shots | Fake Video Shots | Total Number | Fake Audio | Fake Transcript |
|---------|------|------------------|------------------|--------------|------------|-----------------|
| UADFV [57] | 2019 | 49 | 49 | 98 | No | No |
| DeepfakeTIMIT [31] | 2018 | 640 | 320 | 960 | Yes | No |
| FF++ [45] | 2019 | 1,000 | 4,000 | 5,000 | No | No |
| DeeperForensics [28] | 2020 | 50,000 | 10,000 | 60,000 | No | No |
| DFDC [15] | 2020 | 23,654 | 104,500 | 128,154 | Yes | No |
| Celeb-DF [34] | 2020 | 590 | 5,639 | 6,229 | No | No |
| WildDeepfake [63] | 2020 | 3,805 | 3,509 | 7,314 | No | No |
| FakeAVCeleb [30] | 2021 | 500 | 19,500 | 20,000 | Yes | No |
| ForgeryNet [23] | 2021 | 99,630 | 121,617 | 22,1247 | No | No |
| LAV-DF [5] | 2022 | 36,431 | 99,873 | 136,304 | Yes | Yes |
| DF-Platter [38] | 2023 | 764 | 132,496 | 133,260 | No | No |
| IDForge (Ours) | 2024 | **79,827**+ 214,438 (*Ref*) | **169,311** | **249,138**+ 214,438 (*Ref*) | Yes | Yes |

modeling video and audio modalities. Cheng [8] introduced a two-stage method with a face-audio matching pre-training task. Shao *et al.* [48] proposed a method for image-text pair manipulation detection that aligns and aggregates multimodal embeddings. Haliassos *et al.* [22] also put forth a two-stage student-teacher framework. However, the model is fine-tuned on a visual-only forgery detection task, a decision limited by the fact that the datasets they used do not include the accompanying audio with the videos.

In addition, several works have recognized the potential of identity cues in media forgery detection and aim to mine and utilize these cues. Dong *et al.* [17] leverages attributes as prior knowledge associated with the identity to learn identity embeddings. This method is developed using a private, custom visual manipulation dataset to address the aforementioned dataset problems. Cozzolino *et al.* [13] proposed an identity-aware detection approach that trains a 3D morphable model to learn facial representation in an adversarial fashion. The reference videos, whose presentations are compared with that of suspect videos via a fixed threshold, are introduced in the detection stage only. Dong *et al.* [16] introduced an identity consistency transformer that learns identity embedding for both the inner and outer regions of the face. Cozzolino *et al.* [12] proposed an audio-visual media forgery detection method in which audio and visual encoders are trained on real videos to learn person-of-interest representation. Huang et al. [25] proposed to mine both the explicit and implicit identities in suspect videos by applying a novel explicit identity contrast loss and an implicit identity exploration loss.

Though existing methods have made substantial contributions to the field of media forgery detection, they are often constrained by the limitations in current media forgery datasets, as discussed earlier. Therefore, we introduce the identity-driven multimedia dataset IDForge. IDForge not only offers high-quality multimedia forgeries but also includes a complementary reference set that provides abundant identity information, addressing the shortcomings highlighted in recent research.

## 3 IDFORGE DATASET

IDForge contains 54 English-speaking celebrities chosen from the spheres of politics and entertainment. These celebrities are selected due to their fame and media presence on the Internet, which place

them at a high risk of being targeted by forgers. By restricting our subjects to selected celebrities, we can concentrate on creating individual-targeted media forgeries while maintaining high quality. To acquire a more comprehensive representation of each celebrity, we collect 10-30 high-quality, non-overlapping full-length videos for each identity from YouTube, which amount to 847.8 hours.

For developing and testing forgery detectors, we manually select 6 full-length videos for each identity based on the video resolution and clarity of the audio. The remaining videos constitute the reference set. Notably, most videos in existing media forgery datasets have a resolution lower than $480 \times 360$ pixels. For IDForge dataset, we select quality videos with a resolution of $1280 \times 720$ pixels.

Moreover, most full-length videos collected from YouTube have long durations (i.e., $0.5 - 2$ hours) and comprise multiple scenes. We extract video shots showing speakers talking directly to the camera as they reveal aspects of the celebrities' identities and are common targets forgery. We employ a two-step process to preserve these talking scenes. Firstly, the original videos are split into chunks based on scene changes. Each chunk undergoes manual review and is removed if it contains irrelevant scenes. Following this, the remaining chunks are subdivided into smaller video shots at points of silence, yielding sentence-level video shots, each shorter than 20 seconds. However, we observe that some video shots are too brief, lasting less than 3 seconds, due to speakers often pausing within sentences to sustain a proper speaking rhythm, regardless of punctuation. Instead of removing the short video shots, we merge them with the surrounding video shots. Specifically, for a video shot with a duration shorter than 5 seconds, we combine it with its preceding and following clips. This procedure ensures the maximum semantic integrity of the cutout sentences. We obtain $79,827$ pristine video shots following the above process.

### 3.1 Pristine Data Colletion

### 3.2 Individual-targeted Manipulation

**Transcript**. Transcripts, representing the spoken content of the videos, are manipulated with two methods: the LLM and text shuffling. For each pristine video shot, we perform speech recognition to obtain its transcript by Whisper [44]. Instead of simply replacing words in previous works [5, 48], we employ GPT-3.5 [4] to generate

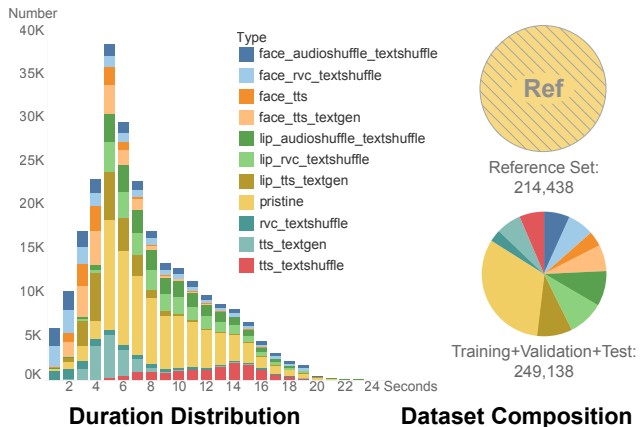

**Figure 2: Statistical information of the IDForge dataset.**

new sentences with similar stylistic properties but distinct semantics. This approach simulates potential LLM misuse in spreading false information. Additionally, to diversify the data, we use text shuffling, which swaps one individual's transcript for another's, resulting in fabricated texts produced initially by humans.

**Audio.** We manipulate audio sequences using TorToiSe [3], RVC [46], and audio shuffling. For each individual, 300-700 audio sequences are collected from pristine video shots to capture the individual's voice and tone. Using these sequences, we produce new ones that reflect celebrities' voices with TorToiSe and RVC. Additionally, we employ audio shuffling to exchange the audio of one individual with that of another of the same gender, similar to the approach used in dubbed videos found online.

**Video.** We use three face-swapping methods: InsightFace [26, 47], SimSwap [7], and InfoSwap [18]. In addition, we utilize an adapted version of the Wav2Lip [41] model for lip-syncing, which has been specifically trained and fine-tuned to handle high-resolution data. For each individual, we collect three frontal images and videos from lookalikes. Each video undergoes random face-swapping with a selected target face. Using Wav2Lip, we adjust the lip movement in video shots to align with the generated audio, achieving synchronized lip movement in video shots.

Besides, since real-world forgeries often involve multiple manipulations across different modalities, we categorize the forgery techniques into seven categories. By using these techniques in different combinations, we obtain 11 distinct types of multimedia forgery. Further details can be found in supplementary materials. For each instance in IDForge, we provide a binary real-fake label and a fine-grained multi-label, indicating whether each forgery method is used.

### 3.3 Statistic and Comparisions

**Dataset Statistic.** Finally, we obtain a total of 169, 311 fake video shots. For these 249,138 video shots, we split them into training, testing, and validation sets, making sure that the video shots in three splits come from different source videos. After splitting, the distribution of video shots across the datasets is as follows: 61.83% are allocated to the training set, 6.95% to the validation set, and the remaining 31.22% to the test set. In total, the constructed dataset

contains 404,319 video shots, and among them, 214,438 pristine video shots belong to the reference set, providing abundant identity information for identity forgery detection. Table 1 shows that IDForge is larger than any previous works. The corresponding proportions of each component are illustrated in Figure 2. It can be observed that all types of forgeries exhibit similar proportions, which are evenly distributed across different durations, with the majority concentrated in 5-7 seconds.

To guide further investigation, we visualize the distribution of features extracted from pre-trained models across three modalities in Figure 3 using T-SNE. As can be seen, the features of both real and fake data are well-mixed within these modalities. Intuitively, visual information could be the most effective way of characterizing the identity. Hence, we further visualize the distribution of visual features, and for better visualization, we color them according to three randomly chosen identities: *id01*, *id11*, and *id21*. The result suggests forgeries in IDForge retain false identities, as these identities reside in distinct clusters, yet these clusters consist of mixed media forgery.

**Comparision with Existing Forgery Datasets.** Table 1 summarizes the existing popular media forgery datasets. Most of them focus on visual manipulation. Among them, DFDC [15], FakeAVCeleb [30], and LAV-DF[5] are most similar to IDForge, as they contain more than just visual manipulation. Although DFDC has few video shots with forged audio, no corresponding labels indicate whether the audio is manipulated, which makes it difficult for forgery detectors to leverage such information. Despite FakeAVCeleb and LAV-DF involving visual and audio manipulation, the cloned voices are of low quality, often characterized by a lack of emotional range. This deficiency in voice quality can be attributed to the inadequate training data for the text-to-speech model (SV2TTS [27]). The FakeAVCeleb dataset comprises 500 subjects, while the LAV-DF contains 153 subjects. Accurately cloning the voice of such many individuals is quite challenging. Furthermore, since fake audio segments are directly integrated into genuine audio sequences in LAV-DF, this may result in noticeable inconsistencies in the audio. Therefore, we limit the number of subjects and train a dedicated voice model for each subject using state-of-the-art methods. LAV-DF is the only dataset that manipulates video transcripts and alters their semantics. However, manipulation in LAV-DF is achieved through word-level replacements, often leading to false transcripts that exhibit noticeable inconsistencies in the context. In IDForge, we use an LLM to generate false transcripts at the sentence level, which results in more natural and contextually appropriate alterations.

### 3.4 User Study on Media Quality

We further conduct a user study to measure the quality of the generated media in IDForge. We randomly sample 3 manipulated video shots from each of the following multimedia forgery datasets: DFDC, FakeAVCeleb, LAV-DF, and IDForge. Additionally, we include 3 video shots from pristine videos and 3 recent multimedia forgeries found online to ensure a comprehensive evaluation.

We develop a website for this user study and the snapshots of the website are in the supplementary material. Then, 31 participants with backgrounds in computer vision are asked to conduct the

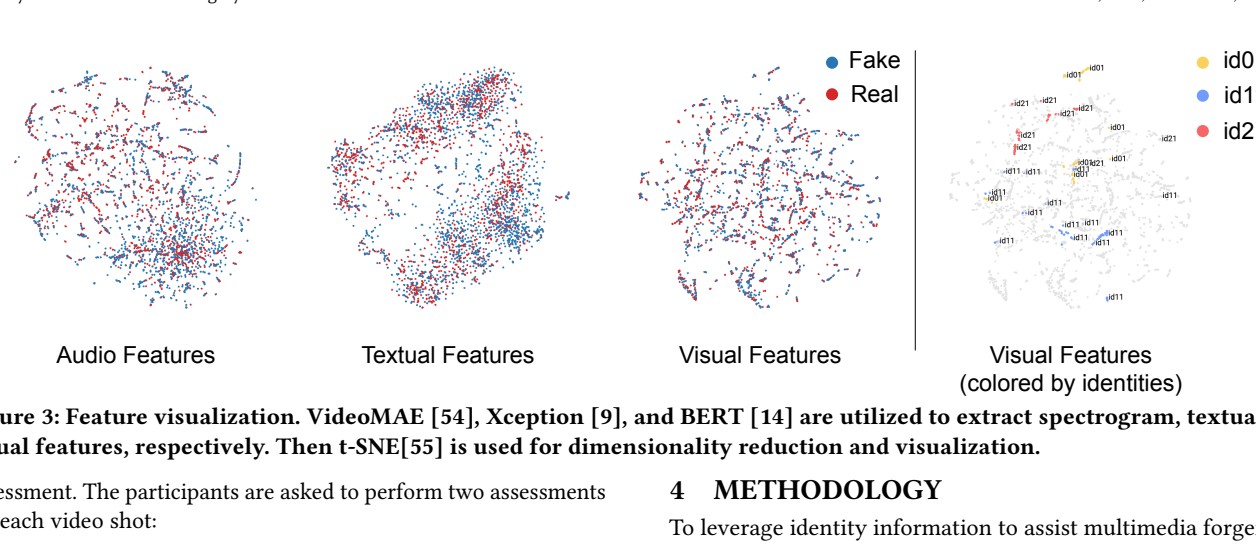

**Figure 3: Feature visualization. VideoMAE [54], Xception [9], and BERT [14] are utilized to extract spectrogram, textual and visual features, respectively. Then t-SNE[55] is used for dimensionality reduction and visualization.**

assessment. The participants are asked to perform two assessments for each video shot:

- Determine the video shots' authenticity.
- Rate the quality of each shot on a five-point scale.

The results are presented in Figure 4. The detectability rates of forgeries from various datasets are shown on the left side of the chart. IDForge exhibits a detectability rate of 25%. This finding suggests that forgeries from IDForge are more challenging for humans to identify as fakes than those from DFDC, FakeAVCeleb, and LAV-DF, highlighting the practical significance of IDForge.

Notably, the detectability rate of IDForge is lower than that of wild multimedia forgeries, indicating that IDForge closely resembles actual forgeries found on the Internet. The media quality of videos is rated on the right side of the chart. IDForge, wild forgeries, and real videos all achieve similar high scores, indicating that the media quality of IDForge is high in terms of both visual and audio aspects.

## 3.5 Ethics Statement

The Institutional Review Board has approved the construction of the IDForge dataset. Primarily intended for academic research, IDForge dataset complies with YouTube's fair use policy. Access is granted only for academic purposes and is subject to application review. The IDForge dataset, featuring selectively sampled video frames, audio, transcripts, and metadata, significantly minimizes the frame count compared to the original YouTube videos. This careful curation is key to mitigating risks associated with the reconstruction or unauthorized distribution of the complete video content.

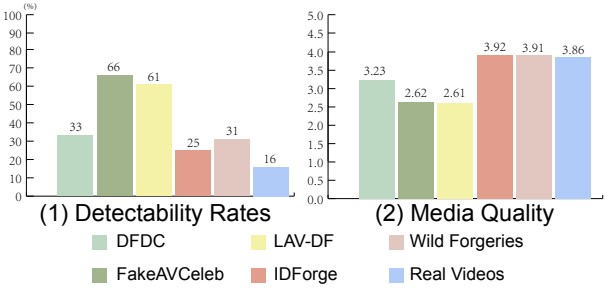

**Figure 4: Results of the user study.**

## 4 METHODOLOGY

To leverage identity information to assist multimedia forgery detection, we propose the **R**eference-Assisted **M**ultimodal **F**orgery **D**etection Network (R-MFDN). As shown in Figure 5, R-MFDN contains three encoders to extract features from frame sequence, audio sequence, and transcript, respectively; then a multimodal fusion module is introduced to fuse the features from different modalities for forgery detection. Identity-aware Contrastive Learning and Cross-Modal Contrastive Learning are introduced in forgery detection to leverage identity information and mine inconsistency between different modalities. It is worth mentioning that the proposed R-MFDN contains two classification heads, one for binary classification and one for multi-label forgery-type classification.

## 4.1 Multi-modal Feature Learning

Given a video shot $X_i = (X_i^v, X_i^a, X_i^t)$ with the label $Y_i = (Y_i^b, Y_i^t)$, where $X_i^v, X_i^a, X_i^t$ denote for its frame sequence, audio sequence, and transcript respectively; $Y_i^b \in \{0, 1\}$ represents whether $X_i$ is a forged video or not, and $Y_i^t \in \{0, 1\}^7$ indicates whether each type of forgery method is applied on $X_i$ or not. For a given video shot $X_i$, R-MFDN first learns its modality-specific features through different encoders.

**Visual encoder.** To capture the temporal inconsistency caused by forgery techniques for visual forgery detection, we follow the frame sampling strategy presented in DIL [21]. That is, for a given frame sequence $X_i^v$, we sample successive frames uniformly to form $n$ frame groups. For each group, the image model $I(\cdot)$ is first applied on each frame to obtain frame-level features. Then, a transformer-based encoder $E_v(\cdot)$ takes the frame feature sequences as input and output group-level features. Average pooling is then applied to obtain clip-level features $\mathbf{f}_i^v$.

**Audio encoder.** Inspired by AST [19], we process an audio sequence $X_i^a$ by initially converting it into mel-spectral features. Next, the generated spectrogram, treated as an image, is partitioned into a sequence of patches. Then a transformer-based encoder $E_a(\cdot)$ is applied on the patches to learn audio feature $\mathbf{f}_i^a$.

**Text encoder.** For a transcript $X_i^t$, we convert it into tokens, then BERT [14] is used to extract the textual feature $\mathbf{f}_i^t$.

**Multimodal feature fusion.** A progressive multimodal feature fusion module is then introduced to fuse visual feature $\mathbf{f}^v$, audio

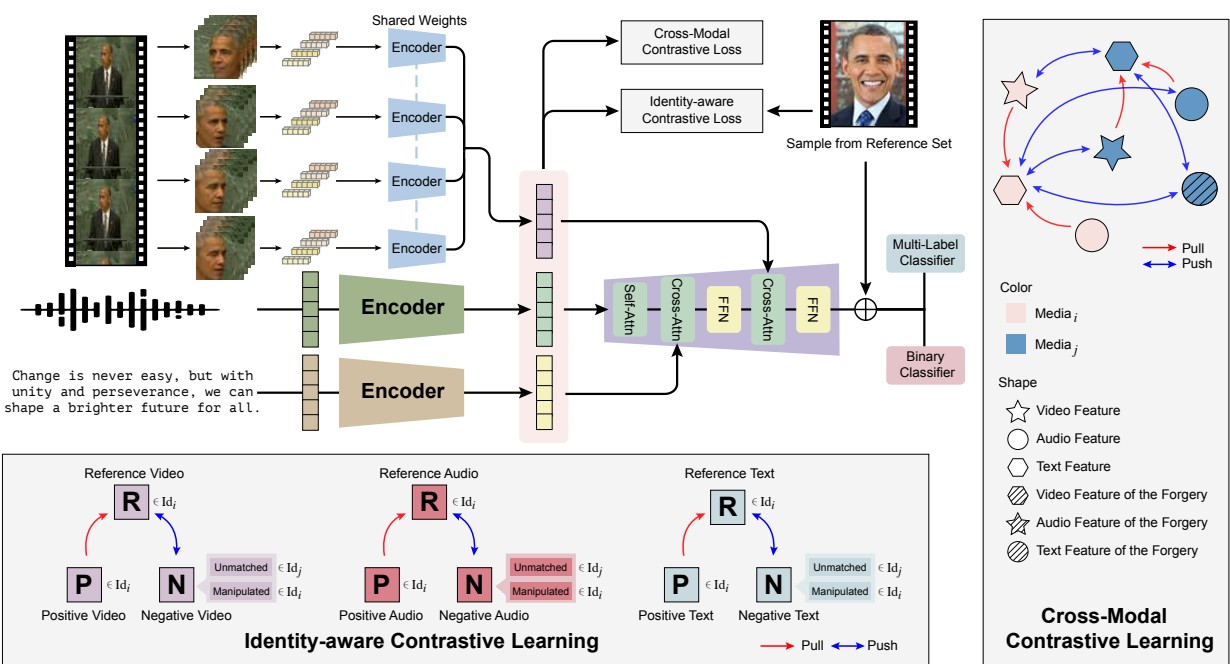

Figure 5: Network overview. The proposed R-MFDN introduces identity-aware contrastive learning to learn identity-sensitive features and captures cross-modal inconsistency through cross-modal contrastive learning.

feature $\mathbf{f}^a$ and textual feature $\mathbf{f}^t$. As shown in Figure 5, the multi-modal feature fusion module first fuses audio and textual features through a cross-attention layer composed of Multihead Attention (MHA) with textual features as query and audio features as key and value.

$$\mathbf{f}^{(a,t)} = \mathrm{MHA}(\mathbf{f}^t, \mathbf{f}^a, \mathbf{f}^a). \tag{1}$$

Before fusion, Self-Attention (SA) is applied to both audio and textual features.

$$\begin{cases} \mathbf{f}^a = \mathbf{f}^a + \mathrm{SA}(\mathbf{f}^a) \\ \mathbf{f}^t = \mathbf{f}^t + \mathrm{SA}(\mathbf{f}^t) \end{cases} \tag{2}$$

Then, the fused features $\mathbf{f}^{(a,t)}$ are forwarded to a feed-forward network (FFN) to further fuse with visual features.

$$\mathbf{f}^{(a,t,v)} = \mathrm{MHA}(\mathrm{FFN}(\mathbf{f}^{(a,t)}), \mathbf{f}^v, \mathbf{f}^v), \tag{3}$$

where $\mathbf{f}^{(a,t,v)}$ are the obtained multi-modal feature.

## 4.2 Cross-modal Contrastive Learning

The existing forgery methods tend to result in inconsistencies across modalities, such as desynchronization between audio and lip movements. To capture such inconsistencies, we introduce a cross-modal contrastive learning strategy. The objective of our cross-modal contrastive learning is to make the distance between paired samples (i.e., Synced media that come from the same pristine video shot) closer than that of the unpaired samples (i.e., unsynced media that come from different pristine video shots or forged videos). We adopt InfoNCE loss for cross-modal contrastive learning.

$$\mathcal{L}_{modal} = -\log \frac{\exp\left(\sigma\left(\mathbf{f}_i^p, \mathbf{f}_i^q\right)/\tau\right)}{\sum_{j=1}^{K} \exp\left(\sigma\left(\mathbf{f}_i^p, \mathbf{f}_j^q\right)/\tau\right)}, \tag{4}$$

where $\mathbf{f}_i^p$ represents the feature for $p$ modality from $i^{th}$ sample, and $\{\mathbf{f}^p, \mathbf{f}^q\} \in \{\{\mathbf{f}^a, \mathbf{f}^v\}, \{\mathbf{f}^a, \mathbf{f}^t\}, \{\mathbf{f}^v, \mathbf{f}^a\}, \{\mathbf{f}^t, \mathbf{f}^a\}\}$; $\tau$ is a temperature hyper-parameter; $K$ denotes the number of sample pairs; $\sigma$ represents the similarity function, and in our implementation, we adopt cosine similarity. To make the encoders more sensitive to inconsistencies in manipulated video shots, we select the features from forged video shots to form negative pairs in contrastive learning.

## 4.3 Identity-aware Contrastive Learning

We introduce identity-aware contrastive learning in the training stage, aiming to learn features sensitive to identities. As shown in Figure 5, for each modality, identity-aware contrastive learning aims to make the distance between features from the same identity closer than that from different identities. Similar to cross-modal contrastive learning, we adopt InfoNCE loss in identity-aware contrastive learning.

Given the $i^{th}$ pristine video shot sample in the training set, its positive pairs are constructed by selecting samples with the same identity in the reference set. Meanwhile, the negative pairs are constructed by selecting samples in the reference set with different identities or selecting manipulated samples in the training set. Hence, we have

$$\mathcal{L}_{identity} = -\log \frac{\exp\left(\sigma\left(\mathbf{f}_i^p, \mathbf{f}_j^p\right)/\tau\right)}{\sum_{k=1}^{K} \exp\left(\sigma\left(\mathbf{f}_i^p, \mathbf{f}_k^p\right)/\tau\right)}, \tag{5}$$

where $\mathbf{f}_i^p$ represents the feature for $p$ modality from $i^{th}$ pristine sample in the training set; the $j^{th}$ sample is sampled from the reference set and with the same identity with $i^{th}$ sample. $\mathbf{f}_k^p$ can be

either the feature from the reference set with different identities or the feature from maniputed training data with the same identity with $i^{th}$ sample. In this way, it enforces the encoders to learn identity-aware features.

## 4.4 Forgery Classification

After obtaining the multimodal feature $\mathbf{f}^{(a,t,v)_i}$ for a given video-shot $X_i$, we fuse it with the identity feature extracted from the reference sample for forgery classification,

$$\mathbf{f}'^{(a,t,v)}_i = \mathbf{f}^{(a,t,v)}_i + \alpha \mathbf{f}^{(a,t,v)}_j, \tag{6}$$

where $\mathbf{f}^{(a,t,v)}_j$ is the multimodal feature for the video-shot $j$ that sampled from the reference set and with the same identity with $i^{th}$ sample; $\alpha$ is a hyper-parameter.

Then two classification heads are introduced for forgery classification; one is the binary classification for forgery detection, denoted as $\Phi_{bic}$, and the other one is the multi-label classification for forgery-techniques prediction, denoted as $\Phi{mlc}$. The loss function for these two classification heads is denoted as $\mathcal{L}_{bic}$ and $\mathcal{L}_{mlc}$, which are as follows.

$$\mathcal{L}_{\text{bic}} = \mathbf{H}(\Phi_{bic}(\mathbf{f}'^{(a,t,v)}_i), Y^b_i), \tag{7}$$

$$\mathcal{L}_{\text{mlc}} = \mathbf{H}(\Phi_{mlc}(\mathbf{f}'^{(a,t,v)}_i), Y^t_i), \tag{8}$$

where $\mathbf{H}(\cdot)$ denotes the cross-entropy function.

**Overall loss function.** The overall loss function for the proposed R-MFDN is given by:

$$\mathcal{L} = \mathcal{L}_{bic} + \mathcal{L}_{mlc} + \beta\mathcal{L}_{modal} + \gamma\mathcal{L}_{identity}, \tag{9}$$

where $\beta$ and $\gamma$ are hyperprameters.

## 5 EXPERIMENTS

### 5.1 Experimental Settings

**Data preprocessing.** We uniformly sample four groups of frames from a video shot, each group containing four frames. We crop the face region of the extracted frames and resize them to $224 \times 224$. We use FFmpeg to extract audio from videos and Whisper to transcribe it. This process provides inputs from three modalities: video, audio, and text.

**Implementation details.** The parameter $\tau$ is set to 1, while $\alpha$ is set to 0.001. Both $\beta$ and $\gamma$ are set to 0.2. We utilize a cosine learning rate scheduler for training. The initial learning rate is set to $5 \times 10^{-5}$. A warm-up phase is incorporated for the first 1000 iteration, during which the learning rate starts from $2 \times 10^{-7}$ and gradually increases to the initial rate. Then, the learning rate follows a cosine decay. The minimum learning rate is constrained to $1 \times 10^{-8}$. We train the networks for 90, 000 iterations on 8 RTX4090 GPUs

**Evaluation Metrics.** Accurate (ACC) and area under the ROC curve (AUC) are used as evaluation metrics for binary forgery detection. For multi-label forgery type prediction, mean average precision (mAP), average per-class F1 (CF1), and average overall F1 (OF1)) are used as evaluation metrics.

### 5.2 Ablation Study

**Effectiveness of Identity-aware Contrastive Learning.** To demonstrate the effect of introducing identity information, we compare the

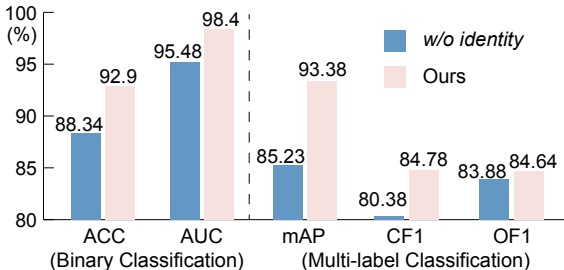

**Figure 6: Comparison between *w/o identity* and R-MFDN.**

performance of our proposed R-MFDN with and without identity-aware contrastive learning. For variant *w/o identity*, the $\mathcal{L}_{identity}$ is removed from the overall loss. Meanwhile, $\alpha$ is set to zero in Eq. 6, since this variant does not use identity information.

Figure 6 shows the comparison results. The R-MFDN model with identity-aware contrastive learning significantly outperforms the variant *w/o identity* across all metrics. The binary classification result shows a 4.56% and a 2.92% performance improvement in accuracy and AUC, respectively. For multi-label classification, improvements are notable, with a 8.15% rise in mAP and 4.4% and 0.76% increases in CF1 and OF1 scores, respectively. These gains underscore the importance of contrasting suspect samples with reference media, enhancing the model's discriminative ability.

**Effectiveness of Multimodal Feature Learning.** To show the advantages of leveraging multimodal features for forgery detection, we compare R-MFDN with the following variants:

- *Visual*: Relies solely on frame sequences for input.
- *Textual*: Relies solely on transcripts for input.
- *Audio*: Relies solely on audio for input.
- *Audio+Textual*: Combines audio and transcripts as inputs.
- *Visual+Audio*: Combines frame sequences and audio as inputs.
- *Visual+Textual*: Combines frame sequences and transcripts as inputs.
- *w/o $\mathcal{L}_{modal}$*: Omits cross-modal contrastive learning by removing the $\mathcal{L}_{modal}$.

Table 2 summarizes the results. From the results, we have the following observations. First, *Audio* performs the best among all the unimodal variants, suggesting that audio's subtle cues like tone and speech patterns are effective for authenticity checks. Second, different modalities play complementary roles to each other. By combining the visual feature with the audio feature, the performances improve from 81.31% to 90.63% in terms of accuracy for binary classification and improve from 83.41% to 86.67% in terms of mAP for forgery type prediction. Third, the proposed cross-modal contrastive learning further boosts the performances for both binary forgery detection and multi-label forgery type prediction, suggesting that cross-modal contrastive learning effectively captures the cross-modal inconsistencies caused by different manipulation techniques, hence improving the final results.

### 5.3 Comparison with State-of-the-Art Methods

**Performance on IDForge**. We further compare the proposed R-MFDN with several state-of-the-art open-sourced forgery detection methods on our IDForge dataset. The methods we compared can be

**Table 2: Ablation study of R-MFDN. The results (%) of the binary classification task are marked in** pink **, while those of the multi-label classification task are in** blue **.**

| Method | ACC | AUC | mAP | CF1 | OF1 |
|---|---|---|---|---|---|
| *Textual* | 67.94 | 74.19 | 58.91 | 46.94 | 51.52 |
| *Visual* | 75.48 | 85.31 | 55.86 | 45.88 | 52.94 |
| *Audio* | 81.31 | 92.31 | 83.41 | 78.47 | 79.15 |
| *Visual+Textual* | 77.67 | 86.46 | 63.75 | 54.92 | 58.31 |
| *Audio+Textual* | 82.66 | 92.72 | 86.14 | 80.28 | 80.24 |
| *Visual+Audio* | 90.63 | 98.12 | 86.67 | 80.84 | 81.82 |
| *w/o* $\mathcal{L}_{modal}$ | 76.39 | 81.49 | 56.53 | 42.56 | 50.02 |
| Ours | **92.90** | **98.40** | **93.38** | **84.78** | **84.64** |

divided into unimodal methods and multimodal methods. Unimodal methods include MesoI4 [1], P3D [43], I3D [6], FTCN [61], LVNet [50], UCF [56], RawNet2 [52], while audio-visual methods include CDCN [29], VFD [8], RealForensics [22]. We add a multi-label classification head to each model, except for VFD, as it detects forgery by comparing the match between face images and voice against a threshold. For both VFD and RealForensics, the reference set of IDForge is used for pre-training.

The results are reported in Table 3. The proposed R-MFDN achieves the best performance among all networks on each evaluation metric. Especially, our network achieves a high accuracy of 92.90% and outperforms RealForensics, VFD, CDCN, and RawNet2 by 3.72%, 6.69%, 13.02%, and 13.69% respectively. The result demonstrated the effectiveness of our R-MFDN in the multimedia forgery detection task, suggesting the advantages of full utilization of multimodal information. Among baseline networks, the multimodal detection methods outperform unimodal methods (e.g., RealForensics is superior to RawNet and UCF for 9.97% and 14.16%on detection accuracy), demonstrating that our dataset is challenging for traditional detection methods which mostly rely on visual information or audio information. In multimodal detection methods, our network still outperforms RealForensics and VFD. One possible reason is that RealForensics and VFD only utilize visual and audio information while overlooking the cues in the transcript and identity. Overall, considering the diversity of manipulation techniques on different modalities in IDForge dataset, we design R-MFDN for multimedia forgery detection, and it gains better performance than other previous methods.

**Performance on FakeAVCeleb**. We further evaluate the performance of our proposed method R-MFDN on the FakeAVCeleb dataset. FakeAVCeleb features cross-modal manipulation and can be expanded with VoxCeleb2 [10] dataset. The source videos for FakeAVCeleb are taken from the VoxCeleb2 dataset, a large-scale speaker recognition collection. Based on the identity names provided in FakeAVCeleb, we gathered additional pristine videos from VoxCeleb2 to build a reference set for identities in FakeAVCeleb.

We also report the performances of state-of-the-art methods on the FakeAVCeleb dataset for comparison, including Joint AV [62], ICT-Ref [16], ID-Reveal [13], and POI-Forensics [12], with scores cited from the original papers. As shown in Table 4, the proposed R-MFDN outperforms existing methods, achieving an accuracy of 96.2% and an AUC of 90.6%.

**Table 3: Performance (%) comparison among open-sourced methods on the IDForge dataset. Methods that rely solely on visual modality, audio modality, and audio-visual modality are marked in** green **,** red **, and** yellow **, respectively.**

| Method | ACC | AUC | mAP | CF1 | OF1 |
|---|---|---|---|---|---|
| MesoI4 [1] | 67.57 | 74.72 | 38.39 | 8.167 | 8.851 |
| P3D [43] | 68.41 | 62.69 | 32.00 | 7.64 | 9.378 |
| I3D [6] | 72.59 | 77.74 | 45.56 | 38.24 | 44.07 |
| FTCN [61] | 75.27 | 79.32 | 58.70 | 51.3 | 56.3 |
| LVNet [50] | 72.60 | 80.52 | 52.77 | 44.12 | 50.16 |
| UCF [56] | 75.02 | 84.49 | 54.56 | 44.73 | 53.21 |
| Ours (*Visual*) | 75.48 | 85.31 | 55.86 | 45.88 | 52.94 |
| RawNet2 [52] | 79.21 | 84.60 | 65.73 | 59.31 | 62.87 |
| Ours (*Audio*) | 81.31 | 92.31 | 83.41 | 78.47 | 79.15 |
| CDCN [29] | 79.88 | 87.43 | 71.17 | 64.07 | 68.21 |
| VFD [8] | 86.21 | 90.70 | - | - | - |
| RealForensics [22] | 89.18 | 93.21 | 88.18 | 80.48 | 80.70 |
| Ours (*Visual+Audio*) | 90.63 | 98.12 | 86.67 | 80.84 | 81.82 |
| Ours | **92.90** | **98.40** | **93.38** | **84.78** | **84.64** |

**Table 4: Performance (%) on the FakeAVCeleb dataset.**

| Method | AUC | ACC |
|---|---|---|
| MesoI4 [1] | 75.8 | 72.2 |
| EfficientNet [53] | - | 81.0 |
| Xception [9] | 76.2 | 71.7 |
| FTCN [61] [61] | 64.9 | 84.0 |
| Joint AV [62] | 48.6 | 55.1 |
| ICT [16] | 63.9 | 68.2 |
| ICT-Ref [16] | 64.5 | 71.9 |
| ID-Reveal [13] | 60.3 | 70.2 |
| POI-Forensics [12] | 86.6 | 94.1 |
| VFD [8] | 81.52 | 86.11 |
| RealForensics [22] | 82.1 | 92.2 |
| Ours | **90.6** | **96.2** |

## 6 CONCLUSION

This paper introduces an identity-driven multimedia forgery dataset IDForge, consisting of 249, 138 high-quality video shots for developing and testing detection methods. IDForge dataset specifically focuses on individual-targeted media forgeries. Audio, video, and text manipulation methods are simultaneously utilized to mimic the speakers' appearances, voices, and speaking styles. Additionally, IDForge provides a reference set containing extra 214, 438 pristine video shots related to the individuals featured in the dataset. We also proposed a novel detection method called R-MFDN, which leverages multimodal and identity information to detect media forgery. Our experimental results demonstrate the effectiveness of the proposed R-MFDN and underscore the potential of identity information. Consequently, IDForge could offer valuable resources in advancing research on harnessing identity information for more effective multimedia forgery detection.

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
