# OpenReview forum: "Identity-Driven Multimedia Forgery Detection via Reference Assistance"
_acmmm.org/ACMMM/2024/Conference — MM2024 Oral_

### Official Review · Reviewer_FgX1 · 2024-05-11

**Rating:** 4
**Confidence:** 3

**Summary:**

This paper proposes to facilitate multimedia video forgery detection with identity information and proposes an identity-driven multimedia forgery dataset IDForge, which contains 463, 576 high-quality video shots. This paper also proposes a simple-yet-effective new model R-MFDN for multimedia forgery detection, which achieves the highest performance.

**Strengths:**

1. The proposed IDForge dataset covers identity information, and is currently the largest multimedia forgery dataset.
2. The proposed R-MFDN is simple-yet-effective and achieves the state-of-the-art performance.
3. The proposed IDForge dataset and the R-MFDN cover vision, audio and text modalities and fit well with the multimedia theme of ACMMM24.

**Limitations:**

1. The proposed IDForge dataset does not make a significant contribution. The previous LAV-DF dataset also covers audio and text modalities.
2. The design of the proposed R-MFDN is straightforward, the novelty is limited.
3. Many previous works on face forgery detection have already proposed to utilize identity information (e.g. ID-Reveal: Identity-Aware DeepFake Video Detection. ICCV2021), the novelty of the motivation is limited.

**Suitability:**

3

---

### Official Review · Reviewer_zQ1W · 2024-05-17

**Rating:** 5
**Confidence:** 4

**Summary:**

This paper points out the two essential aspects in the field of face forgery detection: multimodal content and identity information. Therefore, a large scale dataset IDForge is proposed to assist the research of forgery detection. Moreover, authors also propose a method named RMFDN to achieve the goal of ID-related multimodal detection. Overall, the workload is full and the proposed dataset is benefit for the community.

**Strengths:**

1. The aspects of multimodal content and identity information are important for the face forgery detection
2. The size of proposed dataset is huge and the setting of dataset organization is reasonable
3. The proposed RMFDN achieves great performance in experiment

**Limitations:**

1. Although the identity-related detection are mainly devoted to the face swapping manipulation. In conclusion section, authors still should discuss the detection scenario of other manipulations like face reenactment, even though it is kind of limitation for identity-related method. Authors also can discuss how to improve the face reenactment detection performance in this identity-related task in the furture.
2. It seems that section 3.1 is empty
3. In experiments, it is better to compare with other identity-related methods like ICT[1]
4. I am curious that whether the choice of different reference image will influence the detection performance? It is better to demonstrate it in the experiments.

Ref

[1] Protecting Celebrities with Identity Consistency Transformer

**Suitability:**

3

---

### Official Review · Reviewer_grbb · 2024-05-22

**Rating:** 2
**Confidence:** 3

**Summary:**

This manuscript focuses on detecting multimodal deepfake videos. The author introduces an identity-driven deepfake videos dataset called IDForge, which consists of 463,576 video shots, and develops a Reference-assisted Multimodal Forgery Detection Network (R-MFDN).

**Strengths:**

Strengths:
1. The author has introduced a new multimodal deepfake video dataset (IDForge) to the community, which incorporating richer identity information and combines methods like deepfake videos and deepfake voice to create deepfake videos.
2. The author developed a Reference-assisted Multimodal Forgery Detection Network (R-MFDN) that employs identity-aware contrastive learning to obtain identity-sensitive features and identify cross-modal inconsistencies.

**Limitations:**

Limitations:
1. Usually, "Media Forgery Detection" is a broad topic that includes traditional image tampering detection, video forgery detection, such as using software like Photoshop for forgery. Additionally, new Diffusion-based methods can be used for multimedia forgery and generation, including facial video manipulation. However, the authors only compares methods related to deepfake detection in the manuscript, claiming to outperform state-of-the-art baselines in "media forgery detection," which I find unsuitable.
2. As mentioned above, there are already Diffusion-based facial manipulation methods, e.g., [1], [2]. These types of forgery methods were not included in the dataset, and there was no discussion in the related literature review.
3.  In Section 4.4, how have the hyperparameters of the Loss function been tuned? How sensitive is the algorithm to these parameters? For instance, why was the temperature hyperparameter set to 1 in Identity-aware Contrastive Learning?
4. The dataset consists of 324 video segments, with 79827 clips extracted, averaging over 246 clips per video. Could this setup result in high similarity among these clips, indicating insufficient dataset diversity?
5. For practical deepfake detection scenarios, algorithm generalizability may be more important. I suggest the author conduct experiments on the detection algorithm's generalization abilities, such as training with IDForge and testing on other deepfake datasets.
6. In Table 4, methods such as EfficientNet and Xception are based on single-image inputs, while the approach in this study utilizes multiple images at once. This direct comparison at the end may be considered unfair. It may be worth considering methods like voting to process the results of single-frame or other multi-frame methods before conducting a comparison.
7. Most of the methods compared are from before 2022. I suggest the author explore newer approaches like [3], [4] that have emerged in recent years. It is important for the author's literature review to be more comprehensive.

Based on the reasons mentioned above, I believe that this manuscript still requires more in-depth research. Therefore, I have given it a Weak Reject. I hope that the authors' future improvements will enrich the dataset construction and make the comparative analysis more comprehensive.

[1] Kim, Minchul, Feng Liu, Anil Jain, and Xiaoming Liu. "Dcface: Synthetic face generation with dual condition diffusion model." 2023 IEEE/CVF Conference on Computer Vision and Pattern Recognition (CVPR) (2023): 12715-12725.

[2] Melzi, Pietro, Christian Rathgeb, Rubén Tolosana, Rubén Vera-Rodríguez, Dominik Lawatsch, Florian Domin and Maxim Schaubert. “GANDiffFace: Controllable Generation of Synthetic Datasets for Face Recognition with Realistic Variations.” 2023 IEEE/CVF International Conference on Computer Vision Workshops (ICCVW) (2023): 3078-3087.

[3] Ojha, Utkarsh, Yuheng Li and Yong Jae Lee. “Towards Universal Fake Image Detectors that Generalize Across Generative Models.” 2023 IEEE/CVF Conference on Computer Vision and Pattern Recognition (CVPR) (2023): 24480-24489.

[4] S. Dong, et al., "Implicit Identity Leakage: The Stumbling Block to Improving Deepfake Detection Generalization," 2023 IEEE/CVF Conference on Computer Vision and Pattern Recognition (CVPR) (2023): 3994-4004.

**Suitability:**

3

---

### Official Review · Reviewer_Hxqv · 2024-05-24

**Rating:** 5
**Confidence:** 4

**Summary:**

This study focuses on the detection of video deepfakes and presents the following key contributions:
- The creation of IDForge, a comprehensive dataset comprising 249,138 videos featuring 54 celebrities subjected to manipulation using nine distinct methods across visual, audio, and textual modalities. Notably, the dataset also includes an additional set of 214,438 pristine videos, serving as a reference for identity-based forgery detection.
- Introduction of a novel method named Reference-assisted Multi-modal Forgery Detection Network (R-MFDN), which utilizes contrastive learning and cross-modal contrastive learning techniques.
- Performance evaluation through experiments conducted on both the IDForge and FakeAVCeleb datasets, demonstrating the superior efficacy of the proposed method compared to state-of-the-art approaches.

**Strengths:**

- This work delves into the burgeoning field of AI-enabled media generation and manipulation, addressing a topic of considerable interest.

- The newly introduced dataset enriches the available resources notably by incorporating multimodal manipulations, thus rendering it more reflective of real-world scenarios compared to current datasets. The data collection methods adhere to contemporary standards, and there's commendable utilization of various technologies such as auto-transcription and text manipulation based on large language models, alongside audio manipulation, enhancing the dataset's depth and complexity.

- Additionally, the proposed network Reference-assisted Multi-modal Forgery Detection Network (R-MFDN) outperforms the existing state-of-the-art methods on the proposed dataset, providing a valuable tool for detecting forged media.

**Limitations:**

- The presentation of this paper needs to improve, and there are still typos in the paper. For example, Line 164, Indentity. The authors need carefully check the manuscript.

- The user study in terms of dataset quality is not very clear to me. First, why recruiting 31 participants with background of computer vision? Secondly, the metric of video quality is also not clear. Is this a subjective measure? Or is this measure related to resolution?

- It’s not clear to me what’s the advantage of formulating a deepfake detection as a multi-label classification problem. In my opinion, in real-world applications, people may not care about which manipulation methods are applied. They only care about whether the video is fake or not.

**Suitability:**

3

---

### Meta-Review · Area_Chair_K8QD · 2024-07-03

**Recommendation:** Accept (Oral)
**Confidence:** 5

**Metareview:**

The scope of this work has been over-claimed. It would be better to narrow the scope in the title, e.g., focusing on the deepfake detection, rather than general media forensics. Also, the DM based methods should be discussed, as it is one of the most popular methods of generating fake media. More importantly, the generalization of the proposed method needs to sufficiently studied; otherwise the practical usefulness is rather limited.